# Prevalence of intestinal parasitic infections and genetic differentiation of *Strongyloides stercoralis* among migrant workers from Myanmar, Lao PDR and Cambodia in northeastern Thailand

Wararat Sangwalee[1], Jun Norkaew[1], Sengchoy Inthachak[2], Penchom Janwan[3], Rutchanee Rodpai[4,5], Oranuch Sanpool[4,5], Lakkhana Sadaow[4,5], Patcharaporn Boonroumkaew[4,5], Pewpan M. Intapan[4,5], Wanchai Maleewong[4,5], Tongjit Thanchomnang[6]*

1 Faculty of Public Health, Vongchavalitkul University, Nakhon Ratchasima, Thailand, 2 Faculty of Education, Vongchavalitkul University, Nakhon Ratchasima, Thailand, 3 Department of Medical Technology, School of Allied Health Sciences, Walailak University, Nakhon Si Thammarat, Thailand, 4 Department of Parasitology, Faculty of Medicine, Khon Kaen University, Khon Kaen, Thailand, 5 Mekong Health Science Research Institute, Khon Kaen University, Khon Kaen, Thailand, 6 Faculty of Medicine, Mahasarakham University, Maha Sarakham, Thailand

* tongjit.t@msu.ac.th

## Abstract

Intestinal parasitic infections (IPIs) remain a public-health problem worldwide, including in countries of the Lower Mekong subregion. Increases in human migration from neighboring countries might cause reemerging parasitic infections, leading to spread of parasites in the landscape. Here, we conducted a cross-sectional study to identify the prevalence of IPIs in migrant workers from Myanmar, Lao PDR, and Cambodia who were dwelling in Nakhon Ratchasima Province, northeastern Thailand. The identification of *Strongyloides* species and genetic differentiation of worms from migrant workers with different countries of origin was also assessed. Fresh stool samples were collected from 338 migrant workers and examined for evidence of IPIs using agar plate culture (APC) and the formalin-ethyl acetate concentration technique (FECT). Among those nine samples positive for nematodes by APC, the *Strongyloides* or hookworm species present was confirmed using the polymerase chain reaction (PCR) followed by DNA sequencing. This revealed eight cases of *Strongyloides stercoralis* infection and one of *Necator americanus*. Fifty-one out of 338 individuals (15.09%) were positive for IPIs using FECT and APC. Eggs of *Opisthorchis*-like flukes were the most common parasite (11.83% of samples), followed by *S. stercoralis* (2.37%), *Entamoeba coli* (1.50%), hookworm (0.89%), *Taenia* sp. (0.60%) and *Hymenolepis nana* (0.30%). The genetic differentiation of *S. stercoralis* recovered from migrant workers with different countries of origin was analyzed. Specimens of *S. stercoralis* isolated from workers from Lao PDR, Cambodia and Myanmar were genetically similar to those sequenced from Thailand. However, there were population-genetic differences between *S. stercoralis* from these Southeast Asian countries and other regions of the world. This study demonstrated

**Data Availability Statement:** All relevant data are within the paper and its Supporting Information files.

**Funding:** This research project was financially supported by Mahasarakham University 2021 (TT). a grant from Research Program, Research and Graduate studies, Khon Kaen University (OS, PMI and WM; grant number RP-65-3-001). The funders had no role in study design, data collection and analysis, decision to publish, or preparation of the manuscript.

**Competing interests:** The authors have declared that no competing interests exist.

that IPIs were prevalent in migrant workers in the northeastern region of Thailand. Our findings provided molecular confirmation of the presence of *S. stercoralis* and explored the genetic differentiation of *S. stercoralis* from those infected migrant workers. An effective anti-parasitic drug should be provided for migrant workers and its administration enforced.

## Introduction

Intestinal parasitic infections (IPIs) caused by protozoan and helminth parasites remain a public-health problem worldwide [1–3], including in countries of the Lower Mekong subregion [4–6]. The world's population is increasingly mobile, with many travelling from less-developed to more-developed countries for employment opportunities. Thailand is a middle-income country in which construction, food services, manufacturing, and agricultural sectors have all seen economic expansion. Employment in these sectors often attracts migrant workers. In Thailand, most such workers come from Cambodia, Lao PDR, Myanmar and Vietnam [7]. These countries have all reported a high prevalence of IPIs [8–10]. Infected migrants arriving from these countries may cause reemergence of parasitic infections with the potential to spread across the country. To prevent this, control strategies are urgently required.

In Thailand, the Ministry of Public Health has a health-screening policy with which migrant workers must comply before applying for work. This policy also mandates compulsory administration of 400 mg per kg body weight of albendazole to control IPIs [11] although, as will be seen, this is rarely enforced in practice. Recent studies have reported IPIs in migrant workers in Thailand detected using parasitological methods [12–14]. However, these techniques require an expert microscopist and are inadequate to distinguish between certain parasite species, especially *Strongyloides* spp. [15, 16]. It is necessary to use molecular approaches to confirm the identity of such parasite species [17, 18].

Nematodes of the genus *Strongyloides* are soil-transmitted helminths (STH) which cause strongyloidiasis [19, 20]. The two medically important species of *Strongyloides* are *S. stercoralis* and *S. fuelleborni* [21–24]. Molecular identification of *Strongyloides* spp. is largely based on DNA sequences of mitochondrial (cytochrome C oxidase subunit 1 (*cox*1)) and nuclear ribosomal RNA regions (18S rDNA) [25, 26]. Infection with *S. stercoralis* is more common than with *S. fuelleborni* and can have serious consequences, causing death in immunocompromised patients, those infected with human T-lymphotropic virus 1 (HTLV-1), organ-transplant patients, and those undergoing corticosteroid therapy [27]. In such cases, the parasites can disseminate to various organs of their hosts [28].

Although previous studies using the *cox*1 gene have identified *S. stercoralis* in Thailand by PCR and DNA sequencing [22], migrant workers living in Thailand have not been investigated using this approach. It is possible that molecular studies on the parasites carried by these workers will demonstrate genetic differences related to their country of origin. Understanding the prevalence of IPIs in migrant workers and the extent of genetic differentiation from different geographical origins will be useful for health policies aiming to reduce the transmission of parasites across the country and also to increase understanding of the levels of regional genetic variation [29].

Herein, we aimed to identify IPIs in migrant workers in Nakhon Ratchasima Province, Thailand. We also used *cox*1 DNA and 18S rRNA sequences to confirm the presence of *S. stercoralis* in migrant workers and to evaluate genetic variation in this parasite associated with their country of origin and compared with global geographic differences.

## Materials and methods

### Human ethics statement

This study was approved by the human ethics committees of Mahasarakham University (Human Ethics number: 063-354/2021). Permission was granted by the owners of several factories (eg. food, plastic production, and tapioca starch factories), and by migrants working there who were willing to participate in the current study. All workers recruited to the study completed and signed the written informed-consent forms. These forms had been translated into the language of each migrant worker's home country. At the end of the study, all individuals infected with *S. stercoralis* were treated with a single dose of ivermectin at 20 μg/kg body weight, while other IPIs detected were treated with appropriate anthelminthic drugs.

### Population and sample size calculation

This cross-sectional study was conducted between March 2021 and January 2022 involving four factories in four districts (Muang, Sung Noen, Khon Buri, and Pak Chong Districts) in Nakhon Ratchasima Province, Thailand. This province was chosen because it is one of the most industrialized in northeastern Thailand. A total of 7351 legal migrant workers were living in the province from October 2020 to July 2021, as reported by the migration division of Thailand. These included 4845 migrants from Myanmar, 2010 from Cambodia, and 496 from Lao PDR [7].

The required sample size was calculated using an estimated population proportion formula [30] based on the prevalence of *S. stercoralis* infection at 31.1%, as previously described in Cambodia in year 2017 [31] with 95% confidence interval (z = 1.96), power at 80% (β = 0.20). The required sample size was calculated as 315 individuals. However, a total of 338 participants had returned the informed consent form, so we included all individuals in the study.

### Stool sample collection

Migrant workers were briefed about the objectives of this study by our trained translators before stool samples were collected. After the consent form was obtained and the questionnaire was completed, stool containers were provided to the participants. The process of collecting the stool samples was also explained by our trained translators. Fresh stool samples were collected on the next day in the morning. Agar plate culture (APC) was used to detect *S. stercoralis* and/or hookworm infection [32], and the formalin-ethyl acetate concentration technique (FECT) was used to diagnose other intestinal parasitic infections (protozoans and helminths). Approximately 2 g of fresh stool sample were used for FECT and the protocol was performed in duplicate for each sample as previously described [33]. For the APC method, a total of 4 g of stool sample was plated on the center of an agar culture plate immediately after collection. The agar plates were then sealed with paraffin film and carried to the laboratory at Mahasarakham University. The plates were incubated at room temperature for 4–5 days and examined daily using stereo microscopy. Worms were collected from positive plates. The surface of positive agar plates was washed with 70% alcohol and the washings sedimented by centrifugation at 1500 rpm for 5 min. Some worms were initially examined for their morphological structure under light microscopy after staining with 1% iodine in potassium iodide solution. The remaining worms were kept at -20˚C for DNA extraction and confirmation of parasite species using DNA sequence data [34].

### Data collection

Participants' demographics, socioeconomics, and employment history were obtained using a questionnaire. The questionnaire included both Thai language and translation into languages

used in Myanmar, Cambodia and Lao PDR. Participants completed the questionnaires by themselves.

## DNA extraction, amplification and sequencing

Two worms were taken from each positive culture, individually crushed with a disposable polypropylene pestle (Bellco Glass, Vineland, NJ, USA) and DNA extracted using the Nucleospin Tissue kit (Macherey-Nagel GmbH & Co, Duren, Germany). The protocol was performed according to the manufacturer's instruction. The quality of genomic DNA was checked using a Nanodrop (NanoDrop Technologies, Wilmington, DE, USA). Conventional PCR was performed to amplify an 835-bp region of the mitochondrial cytochrome C oxidase subunit 1 (cox1) gene and nucleotides between positions 843 and 1410 (GenBank accession numbers: AF279916) of the 18S rRNA (hyper-variable region IV (HVR-IV)) gene of *S. stercoralis*. When cox1 and 18S rRNA amplification were negative, a 696-bp portion of the ITS1 and ITS2 regions of the nuclear ribosomal operon was amplified for hookworm detection and identification. PCR conditions and specific primers for *S. stercoralis* and hookworm were used as previously described [34–36]. The PCR products were sequenced using the Applied Biosystems 3730 × I DNA Analyzer and ABI Big Dye Version 3.1 (Applied Biosystems, Foster City, CA), using PCR primers as sequencing primers. All mitochondrial cox1, 18S rRNA and the ITS1 and ITS2 regions of the ribosomal DNA sequences were submitted to the GenBank database and used in BLASTn searches (http://www.ncbi.nlm.nih.gov/).

## Construction of phylogenetic trees

DNA sequences from the mitochondrial cox1 and 18S rRNA and the ITS1 and ITS2 regions of the ribosomal DNA genes were aligned and trimmed using the BioEdit sequence alignment editor program [37]. The phylogenetic trees were constructed using the maximum likelihood method (ML) with 1000 bootstrap re-samplings in the MEGA-X software [38]. The most appropriate substitution model selected was the Hasegawa-Kishino-Yano model [39]. Publicly available sequences of *S. stercoralis*, *S. fuelleborni*, *S. ratti* and hookworms (*Necator americanus* and *Ancylostoma duodenale*) from humans and dogs in different countries were used for reference.

## Genetic differentiation of *S. stercoralis* from migrants from different countries

The pairwise number of differences between cox1 sequences was used to explore population-genetic differentiation of *S. stercoralis* by country of origin of migrant workers (Myanmar, Lao PDR and Cambodia). Published sequences in GenBank from Thailand (n = 58), Myanmar (n = 88), Lao PDR (n = 37) and Cambodia (n = 7) were included. Also included were sequences from China, Japan, Iran, African and South American continents (S1 File). The GenBank database was searched by name of parasite, gene, sequence coverage and country and representative sequences were selected randomly. Population-pairwise genetic differentiation ($F_{ST}$) was determined in Arlequin 3.5.2 software [40]. The standard $F_{ST}$ calculations were tested using 1000 permutations based on the Tajima-Nei model of substitution. An $F_{ST}$ value greater than 0.15 and p-value less than 0.05 can be considered to indicate significant differentiation between populations [41].

## Statistical analyses

Demographics and socioeconomic status of participants were described as numbers and percentages of individuals. Continuous data were described as mean and standard deviation (SD).

Infection with parasites was reported as frequency and percentage. These analyses were conducted using STATA software version 10.1 (Stata Corporation, College Station, TX, USA).

## Results

### Study population and characteristics

We analyzed data from 338 migrant workers who completed and signed the informed-consent forms, completed the questionnaire and provided stool specimens. The participants included 151 migrant workers from Myanmar, 128 from Cambodia, and 59 from Lao PDR. The migrants comprised 184 males (54.44%) and 154 females (45.56%). The average age of participants was 31.76±8.21 years, and the average number of years of living in Thailand was for 6.94 ±4.05 years. Most (87.28%) resided in workers' dormitories and almost none of the workers had ever been treated for helminths (98.22%) (Table 1).

Overall, the prevalence of IPIs according to FECT and APC methods was 15.09% (51/338) as shown in Table 2. *Opisthorchis*-like eggs were the most common infections (11.83%), followed by *S. stercoralis* (2.37%), *Entamoeba coli* (1.50%), hookworm (0.89%), *Taenia* sp (0.60%), and *Hymenolepis nana* (0.30%). Using the FECT alone, 306 samples were examined and 43 individuals (14.05%) were found to be infected with IPIs. Of the 338 stool samples examined using the APC method, nine (9/338; 2.66%) were positive for nematode worms only (3 cases from Lao PDR; 2 cases from Myanmar; and 4 cases from Cambodia) (Table 2). Based on morphological examination of the nematodes, eight samples were identified as *Strongyloides* spp. and one sample was identified as hookworm. Most participants had not undergone stool examination for parasitological diagnosis in the previous year or received any medication for intestinal parasitic infections during their time in Thailand (Table 1 & S1 Table).

### Molecular confirmation

The identity of the nematode parasites found in nine samples by APC was confirmed using DNA sequence data. One nuclear ribosomal sequence was identified as representing *N. americanus* (Fig 1A) and eight *cox*1 (Fig 1B) and 18S rRNA sequences were identified as belonging to *S. stercoralis* (S1 Fig). Our *cox*1 sequences from migrant workers all fell within the type A clade of Nagayasu et al. (2017) [42], in which both human and dog *S. stercoralis* isolates were located. The corresponding GenBank accession numbers are ON954817- ON954824 for *S. stercoralis cox*1 and ON954801- ON954808 for 18S rRNA sequences, while OP002314 is the accession number for the ITS1 and ITS2 regions of the ribosomal DNA region of *N. americanus*.

### Genetic differentiation of *S. stercoralis*

For this analysis, a "population" was regarded as including all *cox*1 sequences from a single country (or region, China, Japan, Iran, African and South American continents). For some analyses, sequences from migrant workers in Thailand were assigned to further "populations" according to their country of origin. The pairwise $F_{ST}$ values indicated varying degrees of genetic differentiation of *S. stercoralis* between nationalities sampled in this study. No sequences of *S. stercoralis* from migrant workers from Lao PDR (3 sequences), Cambodia (3 sequences) and Myanmar (2 sequences) workers were significantly different from those of *S. stercoralis* in Thailand (58 sequences) (Fig 2A). In contrast, *S. stercoralis* sequences (n = 2) from Myanmar migrant workers who lived in Thailand were genetically different from worms from their own country of origin, but not significantly so. When *cox*1 sequences of *S. stercoralis* from Thailand, Lao PDR, Cambodia, and Myanmar were combined into a single

**Table 1. Demographic characteristics and socioeconomic status of the participants ($n$ = 338).**

| Variables | number | Percentage (%) |
|---|---|---|
| **Gender** | | |
| Male | 184 | 54.44 |
| Female | 154 | 45.56 |
| **Age** | | |
| < 35 yrs | 220 | 65.09 |
| ≥35 yrs | 118 | 34.91 |
| Mean ± SD | 31.76±8.21 | |
| **Nationality** | | |
| Myanmar | 151 | 44.67 |
| Cambodia | 128 | 37.87 |
| Lao PDR | 59 | 17.46 |
| **Education** | | |
| Uneducated | 24 | 7.10 |
| Primary school | 184 | 54.44 |
| Secondary school | 128 | 37.87 |
| Bachelor's degree | 2 | 0.59 |
| **Marital status** | | |
| Single | 87 | 25.74 |
| Married | 235 | 69.53 |
| Divorced/widowed | 16 | 4.73 |
| **Current residence** | | |
| Workers' dormitory | 295 | 87.28 |
| House for rent/dormitory | 43 | 12.72 |
| **Previous occupations/positions in country of origin** | | |
| Unemployed | 21 | 6.21 |
| Agriculture | 208 | 61.54 |
| Other | 109 | 32.25 |
| **Length of stay in Thailand** | | |
| <10 yrs | 263 | 77.81 |
| ≥10 yrs | 75 | 22.19 |
| Mean ± SD | 6.94±4.05 | |
| **History of stool examination for helminth eggs in the past one year** | | |
| Yes | 9 | 2.66 |
| No | 329 | 97.34 |
| **History of using drugs for helminth treatment*** | | |
| Yes | 6 | 1.78 |
| No | 332 | 98.22 |

* This refers to time both prior to and following arrival in Thailand.

"population" (Southeast Asia), this population was statistically significantly different from those from other regions (China, Japan, Iran, African and South American continents) (Fig 2B).

## Discussion

In this study, we explored the prevalence of IPIs in migrant workers from Myanmar, Lao PDR and Cambodia who resided in Nakhon Ratchasima Province, northeastern Thailand. We used

**Table 2. Intestinal parasitic infections among 338 immigrant workers in Nakhon Ratchasima Province, northeastern Thailand detected using FECT and APC methods.**

| Parasitic infection | Lao PDR | Myanmar | Cambodia | Total (%) |
|---|---|---|---|---|
| **Using only FECT (*n* = 306)** | | | | |
| No. Examined | 27 | 151 | 128 | 306 |
| No. Infected (%) | 7 (25.93) | 8 (5.30) | 28 (21.88) | 43 (14.05) |
| *Opisthorchis*-like eggs | 6 (22.22) | 6 (3.97) | 28 (21.88) | 40 (13.07) |
| Hookworm | 0 | 2 (1.32) | 1 (0.78) | 3 (0.98) |
| *Entamoeba coli* | 0 | 5 (3.31) | 0 | 5 (1.63) |
| *Taenia* sp. | 2 (7.41) | 0 | 0 | 2 (0.65) |
| *Hymenolepis nana* | 0 | 1 (0.66) | 0 | 1(0.33) |
| *Strongyloides stercoralis* | 0 | 0 | 0 | 0 |
| **Using only APC (*n* = 338)** | | | | |
| No. Examined | 59 | 151 | 128 | 338 |
| No. Infected (%) | 3 (5.08) | 2 (1.32) | 4 (3.13) | 9 (2.66) |
| *Strongyloides stercoralis* | 3 (5.08) | 2 (1.32) | 3 (2.34) | 8 (2.36) |
| Hookworm | 0 | 0 | 1 (0.78) | 1(0.30) |
| **Total (FECT+APC: *n* = 338)** | | | | |
| No. Examined | 59 | 151 | 128 | 338 |
| No. Infected (%) | 10 (16.95) | 10 (6.62) | 31 (24.21) | 51 (15.09) |
| *Opisthorchis*-like eggs | 6 (10.17) | 6 (3.97) | 28 (21.90) | 40 (11.83) |
| Hookworm | 0 | 2 (1.32) | 1 (0.78) | 3 (0.89) |
| *Entamoeba coli* | 0 | 5 (3.31) | 0 | 5 (1.50) |
| *Taenia* sp. | 2 (3.38) | 0 | 0 | 2 (0.60) |
| *Hymenolepis nana* | 0 | 1(0.66) | 0 | 1 (0.30) |
| *Strongyloides stercoralis* | 3 (5.08) | 2 (1.32) | 3 (2.34) | 8 (2.37) |

molecular evidence (mitochondrial *cox*1 gene and 18S rRNA genes) to confirm the identity of *S. stercoralis* and to explore genetic differentiation between samples from workers from different countries. Fifty-one out of 338 individuals (15.09%) were positive for IPIs using FECT and APC methods. The most common parasitic infection was due to *Opisthorchis*-like eggs, followed by *S. stercoralis*, *E. coli*, hookworm, *Taenia* sp., and *H. nana*, in that order. These findings illustrate that IPIs remain prevalent in the Greater Mekong subregion. The finding of *Opisthorchis*-like eggs is of concern. This is a food-borne helminth that is a major risk factor for cholangiocarcinoma in those chronically infected [43]. When using only the APC, nine of 338 participants were positive for nematode larvae. Sequence analysis confirmed that eight of the samples sequenced for *cox*1 and 18S rRNA genes belonged to *S. stercoralis*, while a nuclear ribosomal sequence (ITS1 to ITS2) identified the remaining sample as *N. americanus*. Application of such molecular techniques is an important way to distinguish *S. stercoralis* species and hookworm infections in areas where they overlap.

The overall prevalence of IPIs (as detected by FECT) of 14.05% was lower than reported in previous studies of migrant workers in the same area of Thailand, which found rates of infection to be 24.07% [12] and 27.67% [14]. The prevalence of IPIs of migrant workers in Thailand was lower than in their home countries: Lao PDR (75.8%) [44], Myanmar (85.7%) [45] and Cambodia (50.3%) [46]. The prevalence of *S. stercoralis* infection, in migrant workers was also lower than in the local Thai community, in which prevalence ranges from 23% to 28.9% [15, 47–49]. The low prevalence among migrants might be due to their behavioral patterns or be a result of adequate sanitation in their places of residence. Most workers included in our survey lived in dormitories in the factories where adequate sanitation systems were in place. They

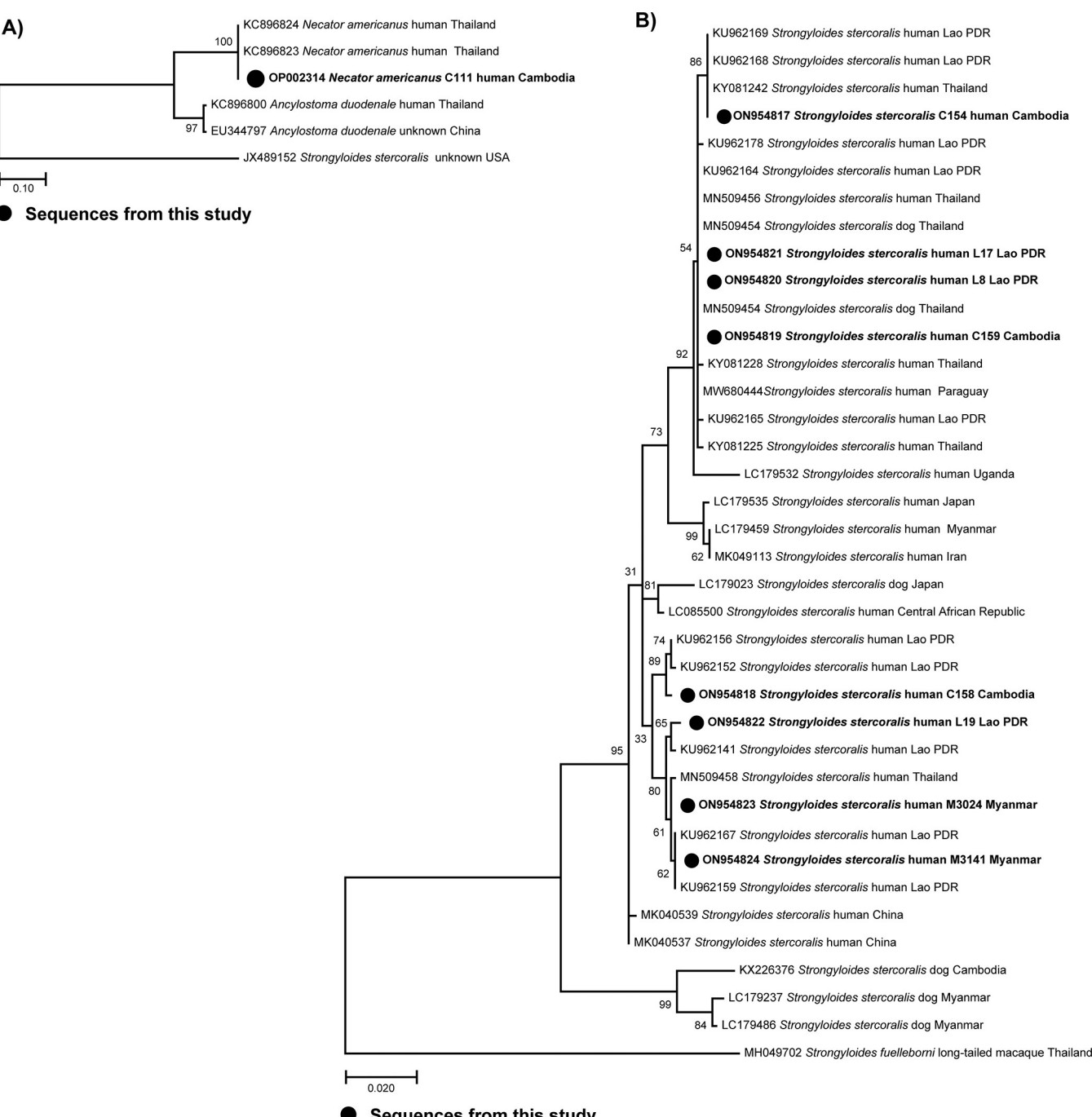

**Fig 1. Phylogenetic tree of *N. americanus* and *S. stercoralis* sequences from legal migrant workers living in Thailand.** A) Phylogenetic position of a ribosomal sequence (ITS1 and ITS2) of *N. americanus* sequence from a migrant worker of Cambodian origin compared with sequences from Thailand. B) Phylogeny based on *cox*1 sequences of *S. stercoralis* from migrant workers in Thailand and reference sequences from GenBank. The sequences obtained from this study (bold letters) are indicated with their accession numbers, hosts, and country.

would have less occupational exposure to IPIs as well as *S. stercoralis* infection than would farmers and fishers in the general Thai community. These findings are consistent with previous reports [12, 14].

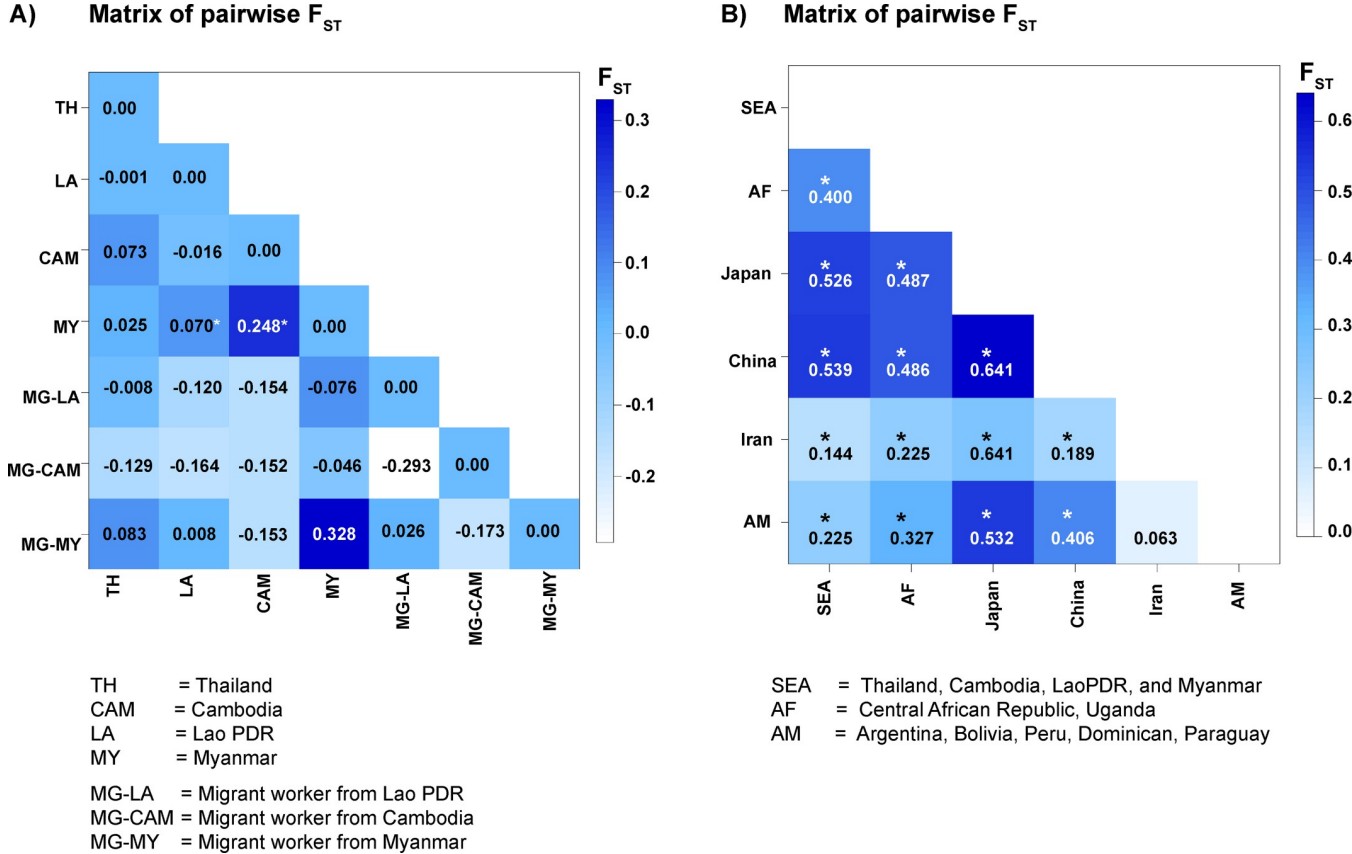

**Fig 2. Population-genetic differentiation between populations (partial *cox*1 gene sequences) of *S. stercoralis* from migrant workers (from Myanmar, Lao PDR and Cambodia) living in Thailand and other parts of the world (China, Japan, Iran, and African and South American continents).** A) Population relatedness represented by pairwise $F_{ST}$ values between worms from each country. B) Population relatedness represented by pairwise $F_{ST}$ values between worms from combined population from Thailand, Myanmar, Lao PDR and Cambodia (Southeast Asia: SEA) and different regions (China, Japan, Iran, African and South American continents). * represents significant difference (*P*<0.005).

In this study, we also assessed the genetic differentiation of *S. stercoralis* samples from infected migrant workers from different countries. Such analysis will shed light on the parasite's origin and spread [50]. Our results from the pairwise $F_{ST}$ values found that sequences of worms isolated from migrants from Lao PDR, Cambodia and Myanmar were similar to those from Thailand, indicating close genetic relationship and high gene flow. This is consistent with a previous report, based on 18S rDNA and *cox*1 sequences, that *S. stercoralis* populations were not very differentiated between countries within the Southeast Asian peninsula [26]. Parasite gene flow is typically strongly influenced by host migration [29]. Given the long span of time through which people have moved throughout Southeast Asia [51, 52], such genetic mixing might be expected. When we extended the analysis to include *cox*1 sequence data from other parts of the world, we found significant differentiation between Southeast Asia (Thailand, Lao PDR, Cambodia and Myanmar) and the other regions (China, Japan, Iran, African and South American continents), supporting geographic restriction of gene flow.

Several parasites, including *S. stercoralis* and *O. viverrini*, can persist in their host for many years. Almost none of the migrant workers had actually been given anthelmintic treatment when (or before) they arrived (Table 1). If they have brought these parasites from their home countries, then the IPI-infected migrant workers found in our study may act as carriers and distributors of parasites which may interrupt the success of IPI control programs in Thailand.

Screening programs, including diagnostic screening and provision of an effective drug treatment, such as ivermectin (for *S. stercoralis*) and praziquantel (for intestinal and liver flukes and tape worms) to alien workers are important for management of IPIs. This will benefit both the government and employers to improve the general health of their workers that will further translate to better productivity and will also reduce the transmission of parasites across the country [53].

The limitations of this study are (a) the small proportion of migrant workers infected with IPIs, (b) stool samples were collected only once from each participant and the quantity was sometimes insufficient for both APC and FECT (thus our figures are likely an underestimate of the prevalence of these parasitic infections) and (c) we cannot be certain whether the infections detected had been acquired in their home countries or after arrival in Thailand. Either scenario is possible given that *S. stercoralis* infections can be long-lived, that almost none of the workers had actually been treated for worms and that there was little or no genetic distinction among the different source and host countries. Furthermore, for genetic analysis of *S. stercoralis*, sequences from migrant workers were too few for robust analysis. In future work, larger samples sizes from migrant workers will be of value. Stool samples should be collected over multiple consecutive days from a greater number of individuals.

## Conclusion

In this work, IPIs were found to be prevalent in migrant workers in the northeastern region of Thailand. The prevalence of *S. stercoralis* infection in migrant workers was low in comparison with the local Thai community. We confirmed the presence of *S. stercoralis* in eight workers and of *N. americanus* in one worker using a molecular approach. Sequences from a portion of the *cox*1 gene were used to explore the levels of genetic differentiation of *S. stercoralis* from migrant workers in Thailand with data from their country of origin. There was significant genetic differentiation between sequences from Southeast Asia and those from other parts of the world but little or no differentiation among the countries represented from Southeast Asia.

## Supporting information

**S1 Table. Details of individuals found to be infected with *S. stercoralis* using APC.**
(PDF)

**S1 Fig. Maximum-likelihood reconstruction of phylogeny based on 18S rRNA sequences (hyper-variable region IV) of *Strongyloides* spp.** Bootstrap scores (percentages of 1000 replications) are presented for each node. The sequences of *Strongyloides* species obtained from GenBank database and this study (bold letters) are indicated with their accession number, hosts, and country. *Necator americanus* was used as an outgroup.
(TIF)

**S1 File. GenBank accession numbers of *cox*1 sequences of *Strongyloides stercoralis* used for analysis in this study.**
(XLSX)

## Acknowledgments

We would like to thank all legal migrant workers in Nakhon Ratchasima Province to participate in this study, and we would like to acknowledge Professor David Blair for the English editing of this manuscript.

## Author Contributions

**Conceptualization:** Wararat Sangwalee, Jun Norkaew, Sengchoy Inthachak, Penchom Janwan, Rutchanee Rodpai, Oranuch Sanpool, Lakkhana Sadaow, Patcharaporn Boonroumkaew, Pewpan M. Intapan, Wanchai Maleewong, Tongjit Thanchomnang.

**Data curation:** Wararat Sangwalee, Jun Norkaew, Sengchoy Inthachak, Oranuch Sanpool, Lakkhana Sadaow.

**Formal analysis:** Wararat Sangwalee, Jun Norkaew, Sengchoy Inthachak, Rutchanee Rodpai, Tongjit Thanchomnang.

**Funding acquisition:** Pewpan M. Intapan, Wanchai Maleewong, Tongjit Thanchomnang.

**Investigation:** Wararat Sangwalee, Penchom Janwan, Rutchanee Rodpai, Oranuch Sanpool, Lakkhana Sadaow, Patcharaporn Boonroumkaew.

**Methodology:** Wararat Sangwalee, Jun Norkaew, Sengchoy Inthachak, Penchom Janwan, Rutchanee Rodpai, Oranuch Sanpool, Lakkhana Sadaow, Patcharaporn Boonroumkaew, Pewpan M. Intapan, Wanchai Maleewong, Tongjit Thanchomnang.

**Project administration:** Tongjit Thanchomnang.

**Supervision:** Pewpan M. Intapan, Wanchai Maleewong, Tongjit Thanchomnang.

**Writing – original draft:** Wararat Sangwalee, Wanchai Maleewong, Tongjit Thanchomnang.

**Writing – review & editing:** Wararat Sangwalee, Jun Norkaew, Sengchoy Inthachak, Penchom Janwan, Rutchanee Rodpai, Oranuch Sanpool, Lakkhana Sadaow, Patcharaporn Boonroumkaew, Pewpan M. Intapan, Wanchai Maleewong, Tongjit Thanchomnang.

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
