## [Decision Letter · Decision Letter 0]

4 Oct 2022

PONE-D-22-24592Prevalence of intestinal parasitic infections and genetic differentiation of Strongyloides stercoralis among migrant workers from Myanmar, Lao PDR and Cambodia in northeastern, ThailandPLOS ONE

Dear Dr. Thanchomnang,

Thank you for submitting your manuscript to PLOS ONE. After careful consideration, we feel that it has merit but does not fully meet PLOS ONE’s publication criteria as it currently stands. Therefore, we invite you to submit a revised version of the manuscript that addresses the points raised during the review process.

We look forward to receiving your revised manuscript.

Kind regards,

Raffi V. Aroian

Academic Editor

PLOS ONE

Journal Requirements:

Additional Editor Comments:

Two expert reviewers and myself have read this manuscript. We agree it is interesting and likely overall of sufficient quality once significant edits and additional information have been included. In particular, reviewer 2 asks for additional sequence comparisons from the data base. Reviewer 1 has also indicated lack of clarity on the statistics. For figure 2b, was there any correction for multiple testing (line 172)? How was this analysis carried out? Both reviewers and myself fiind it very plausible that the infections sequenced came from endogenous sources. This requires clarification in the text and in general a toning down of the text. Upon resubmission, please be sure that all points from both reviewers are addressed.

Reviewers' comments:

Reviewer's Responses to Questions

**Comments to the Author**

1. Is the manuscript technically sound, and do the data support the conclusions?

Reviewer #1: Partly

Reviewer #2: No

2. Has the statistical analysis been performed appropriately and rigorously? 

Reviewer #1: I Don't Know

Reviewer #2: Yes

3. Have the authors made all data underlying the findings in their manuscript fully available?

Reviewer #1: Yes

Reviewer #2: Yes

4. Is the manuscript presented in an intelligible fashion and written in standard English?

Reviewer #1: Yes

Reviewer #2: No

5. Review Comments to the Author

Reviewer #1: In this manuscript the authors surveilled migrant workers in Northern Thailand for intestinal parasites in order to explore if this group of people might cause an influx of parasites from their respective home countries into Thailand. For S. stercoralis, the authors determined cox1 sequences in order to test if the worms in the migrant workers are genetically more similar to the ones in the country of origin of the patient, compared with S. stercoralis found in local people. Due to the low incidence of S. stercoralis in the study population, the genetic study is statistically under powered, a fact that is clearly appreciated by the authors in their discussion. The manuscript is well written and understandable.

While I do find the reported data trustworthy, I disagree with some of their interpretation and presentation. The authors report that the prevalence of S. stercoralis in migrant workers is lower than in the resident population (lines 252,253; it would be interesting to have this comparison also for the other parasites detected in the surveil) and they failed to detect genetic differences between the S. stercoralis in the migrant workers and the local S. stercoralis population. In average, the migrant workers had been living in Thailand for more than six years (line 187). Further, they had presumably received anthelmintic treatment upon their arrival in Thailand (see introduction lines 64,65), although with albendazole, which is not fully active against S. stercoralis. In my opinion, all of this suggests that the majority of the identified infected migrant workers had been infected after their arrival in Thailand. Nevertheless, the authors stress the possibly important role of migrant workers as sources for imported intestinal parasites. This study does not disprove this but it also does not provide any support for it.

Specific points:

Lines 45,46: While technically correct, this sentence towards the end of the abstract is misleading because it suggests a comparably high incidence in migrant workers. The data presented later actually suggest that the prevalence in migrant workers is lower than in local people.

Line 47-49: again, while technically correct, I do not think this statement is appropriate towards the end of the abstract, given that no significant differences were found.

Lines 49,50: While I do agree with this statement in general, this publication does not provide data in support of it. Therefore, this statement is not a conclusion from this study and inappropriate as the last sentence of the abstract.

Lines 167-169: How were the sequences from GenBank to be included in the analysis chosen?

Line 172 (p-value of 0.05): Was there any multiple testing correction included?

Line 187: "working experience" is unclear. From Tab 1 it becomes clear that this is the time the patients had been living in Thailand.

Lines 202,203: Here it is stated that the participants had not received medication against intestinal parasites. However, in lines 64,65 it is explained that such medication is mandatory upon arrival in Thailand.

Paragraph Molecular confirmation (line 208 ...): How many worms per infected patient were sequenced?

Figure 2 B: Please double check the numbers, in particular the white squares. Nei's distance and the pairwise differences between populations should be strongly correlated. For example, it appears weird to me that the pairwise difference between TH and MY is 0 while Nei's difference is close to 1.

Lines260-266: These interpretations assume that the migrant workers had been infected in their home countries. As outlined above, I think infection in Thailand is an, at least equally likely, explanation and needs to be acknowledged and discussed.

Line 292-294: While it is OK and helpful to mention this trend in the results (lines 225,226), repeating it here in Conclusions gives this non-significant finding too much weight.

Lines 294,295: Same comment as lines 49,50. I agree with the statement but it is not a conclusion of this study and therefore inappropriate as the last sentence of the "conclusion" section.

"Conclusions": I think the finding that in this study the prevalence of Strongyloides (and presumably also other intestinal nematodes) was lower in migrant workers compared with the local population would merit mentioning in "conclusions".

Reviewer #2: This manuscript describes a survey of intestinal parasites in immigrant workers in Thailand, and including analysis of the genotypes of Strongyloides stercoralis and hookworm isolates recovered. While of some interest, the absence of HVR-I analysis, and the assumption that all infections were acquired in the worker’s country of origin are flaws in this paper. Furthermore, the isolates sequenced must be compared to isolates from other parts of the world (and shown in the phylogenetic trees) before it can be determined that strains from this region of South-East Asia are very genetically similar and that this is not just due to low genetic diversity in the species sampled at the targeted site.

General comments:

Unless Opisthorchis viverrini was confirmed by morphology or PCR, I recommend referring to “Opisthorchis/Clonorchis spp.”. I would accept noting that Opisthorchis viverrini is the predominant species in this region, but as these are migrants, I do not think that Clonorchis sinensis can be excluded on morphology and geography alone.

Why were haplotype indicative SNPs the Strongyloides stercoralis 18S rRNA HVR-IV region not also examined?

You assume that the hookworm and S. stercoralis infections identified were all acquired in the worker’s country of origin. Given they have received anthelmintic treatment recently, could these hookworm and Strongyloides infections not all be locally acquired Thai S. stercoralis strains? If not, why not? This must be discussed thoroughly and expansively.

Your discussion about genetic relatedness and gene flow in the Thai/Lao/Cambodia/Myanmar region (lines 258-266) is not very meaningful without examples of S. stercoralis with markedly different cox1 haplotypes from other regions of the world. I recommend adding a small number of examples of variant (and similar if found) S. stercoralis Cox1 sequences from the Americas, Europe, Africa and the Australia/Pacific region, as well as China and Japan, and to support that what you are seeing is truly geographically restricted gene flow and not just limited genetic diversity in the species. Please then discuss how the haplotype of your isolates compare to isolates globally.

Could your findings on similarity fo Cox1 sequences in S. stercoralis from your cohort actually reflect low sequence diversity of S. stercoralis at the chosen target? You seem assume that the whole genomes are equally similar as the Cox1 gene sequences. Please discuss, including reference to prior studies which have performed S. stercoralis whole genome sequencing from a single region and compared isolate genomes.

Please add two to three S. stercoralis cox1 lineage B (from dogs in SE Asia and/or Australia) sequences as an additional outlier in your Cox1 tree.

Was Giardia duodenalis considered when viewing the FECT faecal deposits? It is unusual that no Giardia were identified in such a large sample of individuals.

Please make clear in the figure 1 legend that the Genbank accession numbers of your sequences are noted in the tree.

Specific Comments:

Line 75: Please expand this to specifically mention corticosteroid therapy

Line 78: “Using the Cox1 gene” is very non-specific. Do you mean “using a conventional/real-time PCR targeting the Cox1 gene”?

Lines 122-123: How many coverslips of the FECT deposit were screened per sample?

Line 132: Please provide a reference for the conventional PCR used.

Line 161: “Genbank” not “the Genbank”

Line 267-268: As these parasites are already endemic and, in some regions, present in high numbers in Thailand, this argument seems to be redundant.

6. PLOS authors have the option to publish the peer review history of their article (what does this mean?). If published, this will include your full peer review and any attached files.

Reviewer #1: No

Reviewer #2: No

---

## [Author Response · Author response to Decision Letter 0]

18 Nov 2022

Point by point response to reviewer comments

PONE-D-22-24592

Title: Prevalence of intestinal parasitic infections and genetic differentiation of Strongyloides stercoralis among migrant workers from Myanmar, Lao PDR and Cambodia in northeastern Thailand

Additional Editor Comments:

Two expert reviewers and myself have read this manuscript. We agree it is interesting and likely overall of sufficient quality once significant edits and additional information have been included. In particular, reviewer 2 asks for additional sequence comparisons from the data base. 

Reviewer 1 has also indicated lack of clarity on the statistics. For figure 2b, was there any correction for multiple testing (line 172)? How was this analysis carried out? 

Both reviewers and myself find it very plausible that the infections sequenced came from endogenous sources. This requires clarification in the text and in general a toning down of the text. Upon resubmission, please be sure that all points from both reviewers are addressed.

Reply: We would like to thank the editor very much for this positive response. We have added additional sequence comparisons from other geographical regions in Fig. 1 and Fig. 2 as the reviewers suggested. For Fig. 2b, we did not use any multiple-test corrections. The Arlequin program that we used for genetic analysis analyzed the distances between haplotypes without this. We have revised the manuscript and added more content as the reviewers suggested.

Review Comments to the Author

Reviewer #1: In this manuscript the authors surveilled migrant workers in Northern Thailand for intestinal parasites in order to explore if this group of people might cause an influx of parasites from their respective home countries into Thailand. For S. stercoralis, the authors determined cox1 sequences in order to test if the worms in the migrant workers are genetically more similar to the ones in the country of origin of the patient, compared with S. stercoralis found in local people. Due to the low incidence of S. stercoralis in the study population, the genetic study is statistically under powered, a fact that is clearly appreciated by the authors in their discussion. The manuscript is well written and understandable.

While I do find the reported data trustworthy, I disagree with some of their interpretation and presentation. The authors report that the prevalence of S. stercoralis in migrant workers is lower than in the resident population (lines 252,253; it would be interesting to have this comparison also for the other parasites detected in the surveil) and they failed to detect genetic differences between the S. stercoralis in the migrant workers and the local S. stercoralis population. In average, the migrant workers had been living in Thailand for more than six years (line 187). Further, they had presumably received anthelmintic treatment upon their arrival in Thailand (see introduction lines 64,65), although with albendazole, which is not fully active against S. stercoralis. In my opinion, all of this suggests that the majority of the identified infected migrant workers had been infected after their arrival in Thailand. Nevertheless, the authors stress the possibly important role of migrant workers as sources for imported intestinal parasites. This study does not disprove this but it also does not provide any support for it.

Reply: We would like to thank the reviewer very much for these comments out. In fact, almost none of the migrants had been given anthelmintic treatment when they arrived (Table 1), despite government protocols requiring this. For clarity, we have added some sentences in the revised manuscript. First, we added more “although, as will be seen, this is rarely enforced in practice”, please see revised manuscript, page 3, lines 64-65, “and almost none of the workers had ever been treated for helminths (98.22%)”, please see revised manuscript, page 9, lines 195-196. In Table 1, we added as a footnote relating to the question about receiving drugs for helminth treatment* "This refers to time both prior to and following arrival in Thailand", please see revised manuscript, page 10.

Second, we agree with the reviewer’s suggestion that “the immigrants might be infected after their arrival in Thailand” based on cox1 gene sequences in the migrant workers being similar to those of the local S. stercoralis population. We have now acknowledged this as a limitation of our study in the discussion. We have added the sentences “such genetic mixing might be expected. When we extended the analysis to include cox1 sequence data from other parts of the world, we found significant differentiation between Southeast Asia (Thailand, Lao PDR, Cambodia and Myanmar) and the other regions (China, Japan, Iran, African and South American continents), supporting geographic restriction of gene flow.” please see revised manuscript page 15, lines 294-298, and “c) and we cannot be certain whether the infections detected had been acquired in their home countries or after arrival in Thailand. Either scenario is possible given that S. stercoralis infections can be long-lived, that almost none of the workers had actually been treated for worms and that there was little or no genetic distinction among the different source and host countries.”, please see revised manuscript page 16, lines 313-317.

Further, they had presumably received anthelmintic treatment upon their arrival in Thailand (see introduction lines 64,65), although with albendazole, which is not fully active against S. stercoralis.

Reply: We agree with the reviewer’s comment. however, this is the limitation of albendazole treatment. Possibly due to anthelmintic treatment on arrival in Thailand. But in fact, almost none of the migrants had been given anthelmintic treatment when they arrived (Table 1). Table 1 suggests that they had not been given anthelmintic treatment when they arrived, despite government protocols requiring this. We added the sentence “although, as will be seen, this is rarely enforced in practice”, please see revised manuscript, page 3, lines 64-65.

Specific points:

Lines 45,46: While technically correct, this sentence towards the end of the abstract is misleading because it suggests a comparably high incidence in migrant workers. The data presented later actually suggest that the prevalence in migrant workers is lower than in local people.

Reply: We mean that this study has demonstrated that IPIs were still prevalent in migrant workers in the northeastern region of Thailand, not that “the comparison with local people in Thailand”. For clarity, we have added a sentence in the discussion part, “They would have less occupational exposure to IPIs as well as S. stercoralis infection than would farmers and fishers in the general Thai community.”. Please see revised manuscript page 15, lines 283-284.

Line 47-49: again, while technically correct, I do not think this statement is appropriate towards the end of the abstract, given that no significant differences were found.

Reply: We have revised the sentence as follows: “Our findings provided molecular confirmation of the presence of S. stercoralis and explored the genetic differentiation of S. stercoralis from those infected migrant workers. An effective anti-parasitic drug should be provided for migrant workers and its administration enforced.” Please see revised manuscript, page 2-3, lines 46-49. 

Lines 49,50: While I do agree with this statement in general, this publication does not provide data in support of it. Therefore, this statement is not a conclusion from this study and inappropriate as the last sentence of the abstract.

Reply: We have deleted this sentence.

Lines 167-169: How were the sequences from GenBank to be included in the analysis chosen?

Reply: These sequences were selected by the gene position and nucleotide length from each published study. To improve the scientific soundness, we have added more sequence from other countries in the analysis, please see revised sentences, page 8, lines 175-177.

Line 172 (p-value of 0.05): Was there any multiple testing correction included?

Reply: We did not use any multiple-tests corrections. The Arlequin program that we used for genetic analysis has analyzed the distances between haplotypes without such corrections. 

Line 187: "working experience" is unclear. From Tab 1 it becomes clear that this is the time the patients had been living in Thailand.

Reply: We thank the reviewer for your valuable point. We have changed the phrase to “number of years of living in Thailand and Length of stay” here and in Table 1. Please see revised manuscript, page 9, line 194 and Table 1. 

Lines 202,203: Here it is stated that the participants had not received medication against intestinal parasites. However, in lines 64,65 it is explained that such medication is mandatory upon arrival in Thailand.

Reply: We apologize for our lack of clarity. In the Introduction we mean “Thailand has a health-screening policy with which migrant workers must comply before applying for work. This policy also mandates compulsory administration of 400 mg per kg body weight of albendazole to control IPIs. However, according to interview data (Table 1), most participants had not undergone stool examination for parasitological diagnosis in the previous year or received any medication for intestinal parasitic infections during their time in Thailand. Under interview, no immigrant subjects in this study had received anthelmintic treatment. For greater clarity, we have modified Table 1 to “History of using drugs for helminth treatment”, please see Table 1, page 10. For clarity, we have added the sentence “although, as will be seen, this is rarely enforced in practice”, please see revised manuscript, page 3, lines 64-65.

Paragraph Molecular confirmation (line 208 ...): How many worms per infected patient were sequenced?

Reply: We used two worms per infected patient. But we used only one sequence per patient in the analysis because both sequences were always identical. We have added more information in the revised manuscript, please see page 7, lines 142-143.

Figure 2 B: Please double check the numbers, in particular the white squares. Nei's distance and the pairwise differences between populations should be strongly correlated. For example, it appears weird to me that the pairwise difference between TH and MY is 0 while Nei's difference is close to 1.

Reply: We have rechecked and revised Figure 2. For better understanding, we modified figure 2B by combining all Southeast Asia sequences into one “population” and compare with other parts of the world. Please see revised Figure 2B.

Lines260-266: These interpretations assume that the migrant workers had been infected in their home countries. As outlined above, I think infection in Thailand is an, at least equally likely, explanation and needs to be acknowledged and discussed.

Reply: For greater clarity, we have added to the discussion part and limitation part, “please see page 15, lines 288-298 and page 16, lines 313-317.

Line 292-294: While it is OK and helpful to mention this trend in the results (lines 225,226), repeating it here in Conclusions gives this non-significant finding too much weight.

Reply: We have deleted as suggested.

Lines 294,295: Same comment as lines 49,50. I agree with the statement but it is not a conclusion of this study and therefore inappropriate as the last sentence of the "conclusion" section.

Reply: We have deleted the sentence in the abstract and have revised the conclusion.

"Conclusions": I think the finding that in this study the prevalence of Strongyloides (and presumably also other intestinal nematodes) was lower in migrant workers compared with the local population would merit mentioning in "conclusions".

Reply: We have modified the text as the reviewer suggested: “The prevalence of S. stercoralis infection in migrant workers was low in comparison with that in the local Thai community” in the conclusion section. Please see page 16, lines 324-325. Most workers lived in dormitories in the factories where adequate sanitation systems were in place. We have added the sentence in the discussion part, “please see page 15, lines 283-284. 

Finally, we would like to thank the reviewer for the kind and useful suggestions.

Reviewer #2: This manuscript describes a survey of intestinal parasites in immigrant workers in Thailand, and including analysis of the genotypes of Strongyloides stercoralis and hookworm isolates recovered. While of some interest, the absence of HVR-I analysis, and the assumption that all infections were acquired in the worker’s country of origin are flaws in this paper. Furthermore, the isolates sequenced must be compared to isolates from other parts of the world (and shown in the phylogenetic trees) before it can be determined that strains from this region of South-East Asia are very genetically similar and that this is not just due to low genetic diversity in the species sampled at the targeted site.

Reply: We thank the reviewer for these suggestions. No HVR-I sequences were analyzed in our study. However, we have added HVR-IV analysis and included more sequences from other parts of the world as suggested. Please see page 7, lines 148-150 and page 8, lines 160, Materials and Methods section. Result section, page 12, lines 220-224. Discussion section, page 14, line 261 and 270. 

General comments:

Unless Opisthorchis viverrini was confirmed by morphology or PCR, I recommend referring to “Opisthorchis/Clonorchis spp.”. I would accept noting that Opisthorchis viverrini is the predominant species in this region, but as these are migrants, I do not think that Clonorchis sinensis can be excluded on morphology and geography alone.

Reply: Clonorchis sinensis has only once been reported from human populations in Thailand (Traub et al. A new PCR-based approach indicates the range of Clonorchis sinensis now extends to Central Thailand. PLoS Negl Trop Dis. 2009;3(1):e367.), and never from Lao PDR, Cambodia, and Myanmar (the countries from which our subjects had come). Buathong et al. (Molecular discrimination of Opisthorchis-like eggs from residents in a rural community of central Thailand. PLoS Negl Trop Dis. 2017 Nov 2;11(11):e0006030.) reported that Opisthorchis-like eggs from the same area as was sampled by Traub et al. (2009) were O. viverrini according to molecular data. We therefore feel secure in the assumption that the eggs found were from O. viverrini, but have adopted the reviewer’s suggestion and used the phrase “Opisthorchis-like eggs”. 

Why were haplotype indicative SNPs the Strongyloides stercoralis 18S rRNA HVR-IV region not also examined?

Reply: We used the 18S rRNA HVR-IV region for species identification, but it is not suitable for analysis of genetic variation because of its high degree of sequence conservation. However, we have added more data and modified the phylogenetic tree in supplementary Fig1 (S1 Fig).

You assume that the hookworm and S. stercoralis infections identified were all acquired in the worker’s country of origin. Given they have received anthelmintic treatment recently, could this hookworm and Strongyloides infections not all be locally acquired Thai S. stercoralis strains? If not, why not? This must be discussed thoroughly and expansively.

Reply: We apologize to the reviewer for the lack of clarity. Thailand has a health-screening policy with which migrant workers must comply before applying for work. This policy also mandates compulsory administration of 400 mg per kg body weight of albendazole to control IPIs when a positive result was found. In fact, almost none of the migrants had been given anthelmintic treatment when they arrived (Table 1), despite government protocols requiring this. For clarity, we have added some sentences in the revised manuscript. First, we have added “although, as will be seen, this is rarely enforced in practice”, please see revised manuscript, page 3, lines 64-65, “and almost none of the workers had ever been treated for helminths (98.22%)”, please see revised manuscript, page 9, lines 195-196. In the discussion part and limitations, please see revised manuscript, page 15, lines 300-303 and page 16, lines 313-317.

Your discussion about genetic relatedness and gene flow in the Thai/Lao/Cambodia/Myanmar region (lines 258-266) is not very meaningful without examples of S. stercoralis with markedly different cox1 haplotypes from other regions of the world. I recommend adding a small number of examples of variant (and similar if found) S. stercoralis Cox1 sequences from the Americas, Europe, Africa and the Australia/Pacific region, as well as China and Japan, and to support that what you are seeing is truly geographically restricted gene flow and not just limited genetic diversity in the species. Please then discuss how the haplotype of your isolates compare to isolates globally.

Reply: We have added and analyzed more data as the reviewer suggested. Please see page 12-13, lines 235-238, page 13, lines 244-247 and page 15, lines 294-298.

Could your findings on similarity for Cox1 sequences in S. stercoralis from your cohort actually reflect low sequence diversity of S. stercoralis at the chosen target? You seem assume that the whole genomes are equally similar as the Cox1 gene sequences. Please discuss, including reference to prior studies which have performed S. stercoralis whole genome sequencing from a single region and compared isolate genomes.

Reply: We have added more discussion as the reviewer suggested. We have added some the sentences discussion part, please see revised manuscript page 15, lines 288-298.

Please add two to three S. stercoralis cox1 lineage B (from dogs in SE Asia and/or Australia) sequences as an additional outlier in your Cox1 tree.

Reply: We have now done this. Please see the revised Fig1B. We have added the sentence “Our cox1 sequences from migrant workers all fell within the type A clade of Nagayasu et al. (2017), in which both human and dog S. stercoralis isolates were located. Please see page 12, lines 220-224.

Was Giardia duodenalis considered when viewing the FECT faecal deposits? It is unusual that no Giardia were identified in such a large sample of individuals.

Reply: No Giardia stage was found in the present study.

Please make clear in the figure 1 legend that the Genbank accession numbers of your sequences are noted in the tree.

Reply: We have added the sentence “The sequences obtained from this study (shown in bold lettering) are indicated with their accession number, hosts, and country” in Figure 1 legend and in the results, page 12, lines 231-232.

Specific Comments:

Line 75: Please expand this to specifically mention corticosteroid therapy

Reply: We added the words “and those undergoing corticosteroid therapy”, please see revised manuscript, page 4, line 77. 

Line 78: “Using the Cox1 gene” is very non-specific. Do you mean “using a conventional/real-time PCR targeting the Cox1 gene”?

Reply: We have modified the sentence to “Although previous studies using the cox1 gene have identified S. stercoralis in Thailand by PCR and DNA sequencing.” please see revised manuscript, page 4, lines 80. 

Lines 122-123: How many coverslips of the FECT deposit were screened per sample?

Reply: Two coverslips per sample. We have added the sentence “…in duplicate for each sample ….” please see revised manuscript, page 6, lines 124-125. 

Line 132: Please provide a reference for the conventional PCR used.

Reply: We have cited an appropriate reference. Please see revised manuscript, page 6, line 133.

Line 161: “Genbank” not “the Genbank”

Reply: We have modified as suggested. 

Line 267-268: As these parasites are already endemic and, in some regions, present in high numbers in Thailand, this argument seems to be redundant.

Reply: We have deleted this as you suggested.

Finally, we would like to thank you very much. Your comments are encouraging, helpful, and much appreciated.

---

## [Decision Letter · Decision Letter 1]

14 Dec 2022

Prevalence of intestinal parasitic infections and genetic differentiation of Strongyloides stercoralis among migrant workers from Myanmar, Lao PDR and Cambodia in northeastern Thailand

PONE-D-22-24592R1

Dear Dr. Thanchomnang,

We’re pleased to inform you that your manuscript has been judged scientifically suitable for publication and will be formally accepted for publication once it meets all outstanding technical requirements.

Kind regards,

Raffi V. Aroian

Academic Editor

PLOS ONE

Additional Editor Comments (optional):

Reviewers' comments:

Reviewer's Responses to Questions

**Comments to the Author**

1. If the authors have adequately addressed your comments raised in a previous round of review and you feel that this manuscript is now acceptable for publication, you may indicate that here to bypass the “Comments to the Author” section, enter your conflict of interest statement in the “Confidential to Editor” section, and submit your "Accept" recommendation.

Reviewer #1: All comments have been addressed

Reviewer #2: All comments have been addressed

2. Is the manuscript technically sound, and do the data support the conclusions?

Reviewer #1: Yes

Reviewer #2: Yes

3. Has the statistical analysis been performed appropriately and rigorously? 

Reviewer #1: I Don't Know

Reviewer #2: Yes

4. Have the authors made all data underlying the findings in their manuscript fully available?

Reviewer #1: Yes

Reviewer #2: Yes

5. Is the manuscript presented in an intelligible fashion and written in standard English?

Reviewer #1: Yes

Reviewer #2: Yes

6. Review Comments to the Author

Reviewer #1: The authors did respond carefully to the points raised by the reviewers and the editor. I think the manuscript is now acceptable for publication.

Reviewer #2: The authors have satisfactorily addressed all of my review comments. Thank you for your comprehensive response.

7. PLOS authors have the option to publish the peer review history of their article (what does this mean?). If published, this will include your full peer review and any attached files.

Reviewer #1: No

Reviewer #2: No

---

## [Editor Report · Acceptance letter]

22 Dec 2022

PONE-D-22-24592R1 

Prevalence of intestinal parasitic infections and genetic differentiation of *Strongyloides stercoralis* among migrant workers from Myanmar, Lao PDR and Cambodia in northeastern Thailand 

Dear Dr. Thanchomnang:

I'm pleased to inform you that your manuscript has been deemed suitable for publication in PLOS ONE. Congratulations! Your manuscript is now with our production department. 

Kind regards, 

on behalf of

Prof. Raffi V. Aroian 

Academic Editor

PLOS ONE